# A wearable electrostimulation-augmented ionic-gel photothermal patch doped with MXene for skin tumor treatment

Xingkai Ju[1,2], Jiao Kong[1,2], Guohua Qi [1] ✉, Shuping Hou[1,2], Xingkang Diao[1,2], Shaojun Dong [1,2] & Yongdong Jin [1,2,3] ✉

A wearable biological patch capable of producing multiple responses to light and electricity without interfering with daily activities is highly desired for skin cancer treatment, but remains a key challenge. Herein, the skin-mountable electrostimulation-augmented photothermal patch (eT-patch) comprising transparent ionic gel with MXene ($Ti_3C_2T_x$) doping is developed and applied for the treatment of melanoma under photostimulation at 0.5 W/cm². The eT-patch designed has superior photothermal and electrical characteristics owing to ionic gels doped with MXene which provides high photothermal conversion efficiency and electrical conductivity as a medium. Simultaneously, the ionic gel-based eT-patch having excellent optical transparency actualizes real-time observation of skin response and melanoma treatment process under photothermal and electrical stimulation (PES) co-therapy. Systematical cellular study on anti-tumor mechanism of the eT-patch under PES treatment revealed that eT-patch under PES treatment can synergically trigger cancer cell apoptosis and pyroptosis, which together lead to the death of melanoma cells. Due to the obvious advantages of relatively safe and less side effects in healthy organs, the developed eT-patch provides a promising cost-effective therapeutic strategy for skin tumors and will open a new avenue for biomedical applications of ionic gels.

Melanoma is a malignant skin tumor arising in melanocytes, which is easily metastasized and aggressive resulting in a low survival rate[1]. Consequently, a simple, feasible, and high-effective treatment strategy remains a formidable challenge in the field. Smart and wearable biopatches have recently provided a promising and auxiliary approach for skin tumor treatment via external stimulation to trigger irreversible tumor cell damage. So far, a variety of self-powered electrical patches and hyperthermia based on Joule heat have been developed for skin tumor treatment[2,3]. However, most of the reported patches have a complex structure and fabrication process with high requirements for equipment and preparation difficulty, making their preparation costly and time consuming. These shortcomings limit their potential clinical applications, so that the development of a simple, multi-responsive and wearable patch is urgently needed for effective melanoma treatment.

Ionic gels as flexible materials have a similar structure to hydrogels and are usually made by mixing organic polymers with salt electrolyte materials that can be electrolyzed as ions[4]. Ionic liquids disperse in the ionic gel framework structure, which renders them some additional or superior properties such as higher ionic

[1]State Key Laboratory of Electroanalytical Chemistry, Changchun Institute of Applied Chemistry, Chinese Academy of Sciences, Changchun 130022, China. [2]School of Applied Chemistry and Engineering, University of Science and Technology of China, Hefei 230026, China. [3]Guangdong Key Laboratory of Biomedical Measurements and Ultrasound Imaging, School of Biomedical Engineering, Shenzhen University Medical School, Shenzhen University, Shenzhen 518060, China. ✉e-mail: ghqi@ciac.ac.cn; ydjin@ciac.ac.cn

conductivity[5], electrochemical and thermal stability[6] and better anti-bacterial ability[7]. Modulating an ionic gel framework is a feasible means to obtain ionic gel patches with certain optical transparency, electrical conductivity, and photothermal properties[8]. Although ionic gel is a good candidate for a wearable patch, to enhance its photo-electric response to boost photothermal conversion and electrical stimulation (ES) efficacy for effective skin tumor treatment, the doping of photothermal agents is often required.

MXene, as a class of versatile two-dimensional (2D) layered materials, has become the research hotspot due to superior features such as simple preparation, good electroconductivity, photothermal properties and so on[9]. Doping ionic gel with MXene would be a good excellentution to obtain patches with improved photothermal properties and could further enhance tensile property as well as electrical conductivity, which would be a promising platform to trigger ES and photothermal treatment (PTT) for tumor treatment during photo-electric co-stimulation. Importantly, ES is an effective physical modulator of cellular activity with the advantages of little damage, low induction of immune response, and repeatable operation[10]. Compared to other mechanical or chemical stimulations, ES has been proven an excellent technique to regulate cell migration[11], proliferation[12], differentiation[13], and death[14] for applications in wound healing[15], neurological recovery[16] and especially cancer therapy[17,18]. And ES induces mitochondrial dysfunction, which ultimately leads to oxygen radical storm production, which leads to disruption of cellular redox homeostasis and DNA damage, ultimately leading to cell death[19]. ES with patches can achieve full coverage of the entire tumor, avoiding the risk of tumor cells escaping during treatment[20]. Therefore, the combined use of ES with other therapeutic methods, such as PTT, which has been extensively developed for tumor treatment due to its low invasiveness and minimal tissue damage[21], would be promising for achieving high-efficacy tumor treatment. The PTT relies on photosensitizers to absorb incident light and convert the absorbed photon energy into heat, resulting in a rapid increase of local cell temperature over a certain period of time and the destruction of tumors at high temperatures[22,23]. However, traditional photothermal materials, which are mainly made of metal nanomaterials, are often difficult to make close contact with the skin, and the condition of the skin beneath the materials with poor transparency cannot be visually detected in real-time during treatment, which may lead to overheating to skin burns.

In this work, we developed a wearable biological electrothermal patch (eT-patch) based on transparent ionic gel with MXene doping, with improved electrical conductivity and photothermal properties for biomedical applications. The developed eT-patch possesses good optical transparency to actualize real-time visual inspection of the treatment effect of melanoma under the PTT and ES (PES) co-treatment. Significant tumor suppression was affirmed after the PES treatment, and the anti-tumor mechanisms of the eT-patch were revealed from cell levels, as schematically shown in Fig. 1. Since the PES treatment is relatively safe, as no evident damage to the main organs of mice transplanted with melanoma, the eT-patch has potential clinical applications for skin tumor treatment.

## Results

### Preparation and characterization of eT-patch

Firstly, the MXene nanosheets of $Ti_3C_2T_x$ were obtained by etching $Ti_3AlC_2$ powder (MAX) in a mixed solution of lithium fluoride and hydrochloric acid, then centrifuged and dispersed into water to obtain nanosheets of $Ti_3C_2T_x$ colloids, and after freeze-drying to obtain MXene powder. As shown in Fig. 2a, b, the ultrathin 2D layered structure of the sample was clearly observed from the scanning electron microscopy (SEM), and also transmission electron microscopy (TEM) imaging using nanopore-arrayed anodic alumina as a support, indicating successful preparation of the chemically exfoliated $Ti_3C_2T_x$ nanosheets. The thickness of the $Ti_3C_2T_x$ nanosheets measured by atomic force microscopy is about 2 nm, as shown in Fig. 2c, which is consistent with the work reported previously[24]. Subsequently, the crystal structure of $Ti_3C_2T_x$ nanosheets was measured using X-ray diffraction (XRD). As shown the XRD pattern of MAX powder with $Ti_3C_2T_x$ in Fig. 2d, the (002) peak distinctly shifted from 9.5° to a smaller angle of 6.5° and the three peaks assigned to (101), (104), and (105) associated with MAX were disappeared, due to the transformation of $Ti_3C_2T_x$ from $Ti_3AlC_2$ to achieve the intercalation, and the introduction to the surface of -O, -OH and -F end groups[25]. To obtain further information on the surface groups of $Ti_3C_2T_x$ nanosheets, they were detected by X-ray photoelectron spectroscopy (XPS). The XPS spectrum of the nanosheets shows typical peaks of Ti2$p$, C1$s$, O1$s$, and F1$s$ from 0 to 1000 eV (Supplementary Fig. 1), the peaks of which were located at 33, 284, 454, 531, 563, 682, 827, and 976 eV, respectively. The result confirmed that the surface of nanosheets prepared contains

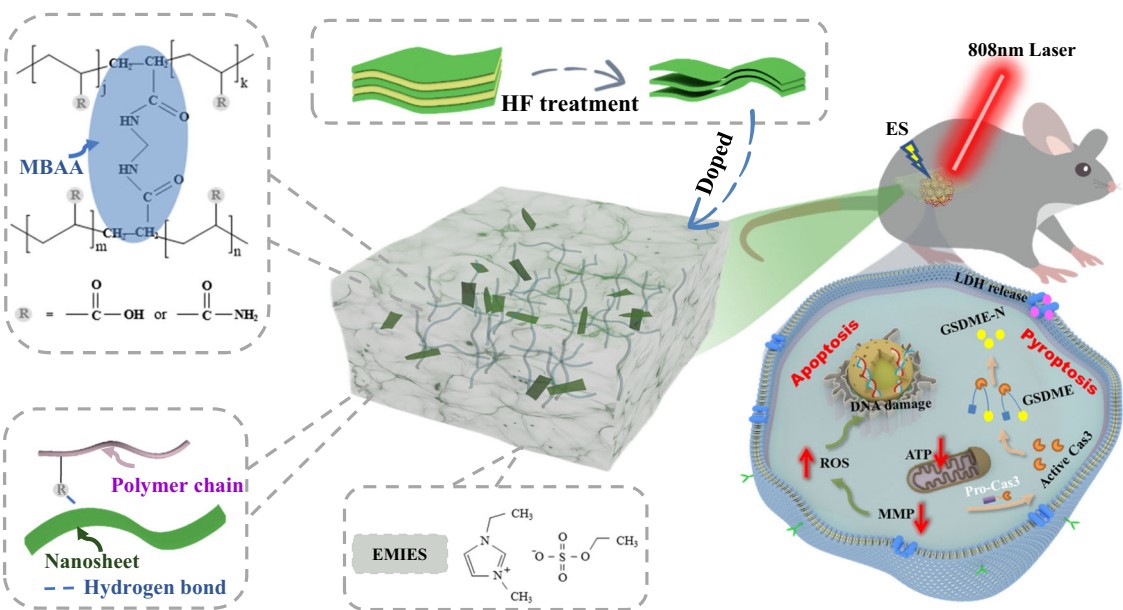

**Fig. 1 | Ionic gel based eT-patch for subcutaneous tumor therapy.** Schematic illustration of the biological eT-patch prepared from ionic gel doped with MXene and their application for skin tumor therapy under synergistic photothermal and electrical stimulation.

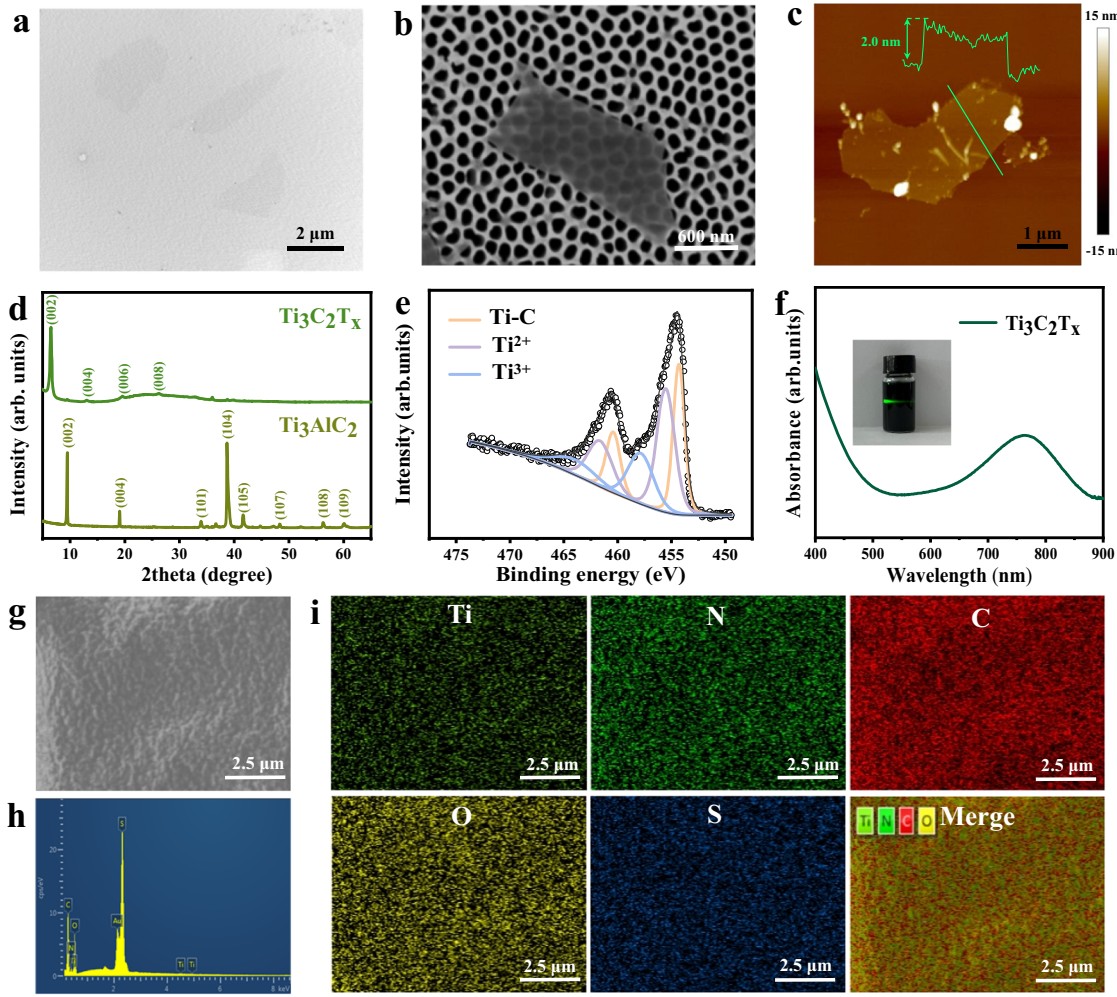

**Fig. 2 | Characterizations of photothermal agents MXene and ionic gels.** TEM image (**a**), SEM image (**b**), and AFM image and line-scan height analysis (**c**) of the $Ti_3C_2T_x$, respectively. **d** X-ray diffraction of the $Ti_3AlC_2$ and $Ti_3C_2T_x$. **e** High-resolution X-ray photoelectron spectroscopy spectra of the Ti 2p region of $Ti_3C_2T_x$ nanoflakes. **f** UV-Vis absorbance spectrum of the liquid exfoliated $Ti_3C_2T_x$ nanoflakes dispersion solution. **g** SEM image of the MXene doped ionic gels. **h** Energy-dispersive spectroscopy of the ionic gels. **i** SEM elemental mapping analyses of the eT-patch. The experiments were repeated for three times with similar results obtained.

mainly surface end groups of -O, -OH, and -F[26]. Figure 2e shows the Ti 2$p$ spectrum of $Ti_3C_2T_x$ nanosheets, while 454.3 and 460.5 eV correspond to the Ti-C bond. The peaks at 455.5 and 461.6 eV are mainly attributed to $Ti^{2+}$, while the peaks of $Ti^{3+}$ were located at 457.8 and 464.3 eV and no peak is detected at 488.8 eV, indicating that no oxidation of $Ti_3C_2T_x$ nanosheets occurred during the delamination process[27,28]. The $Ti_3C_2T_x$ nanosheets obtained by etching can be well dispersed in water and exhibit a significant Tyndall effect (as shown in the inset of Fig. 2f). Meanwhile, the optical properties of the colloidal $Ti_3C_2T_x$ nanosheets in water were examined using UV absorption spectroscopy (Fig. 2f) and observed a significant absorption peak at 768 nm, which can be coupled very well with the 808 nm laser for subsequent PTT of cancer. In this work, we selected B16F10 cells for the next experiments which are derived from C57BL/6J mouse spontaneous tumor cells. The biocompatibility of $Ti_3C_2T_x$ nanosheets was further checked using the standardized MTT assay (Supplementary Fig. 2), the result of which manifested the good cell viability of B16F10 cells after incubated with 45 μg/mL of the nanosheets. The evident temperature elevation of MXene medium irradiated by 808 nm laser was observed, compared with pure $H_2O$ (Supplementary Fig. 3). The cell viability of B16F10 cells incubated with MXene was gradually reduced with laser power boosted (Supplementary Fig. 4). Moreover, the photothermal conversion efficiency of $Ti_3C_2T_x$ nanosheets was

calculated to be ~30% (Supplementary Fig. 5), higher than that of Au nanorods (21%)[29] and $Cu_9S_5NCs$ (25.7%)[30]. Therefore, the MXene nanosheet as an ideal photothermal agent was used for doping in ionic gel patches in the following experiments.

The ionic gel patches doped with MXene nanosheets were then prepared by ultraviolet lamp polymerization in a mold (Supplementary Fig. 6). The relatively flat surface and internal porous lamellar structure of the patch were observed from SEM images, as displayed in Fig. 2g and Supplementary Fig. 7. The SEM energy spectrum and element mapping clearly showed uniform dispersion of Ti element in the ionic gel (Fig. 2h, i), indicating the successful doping of MXene in ionic gel. Due to the interaction with the polymer network to form hydrogen bonding after the addition of $Ti_3C_2T_x$, the resultant composite patches possess great ductility after the $Ti_3C_2T_x$ doping, compared with patches made of pure ionic gels. The stress-strain curves of the patches were tested by a universal tester (Fig. 3a and Supplementary Fig. 8). As seen from Fig. 3b, c, the elongation at break and toughness of the ionic gels boosted with increasing the doping content of $Ti_3C_2T_x$. The elongation at break (at 5.75 MPa) of ionic gel doped with 0.8 mg/mL $Ti_3C_2T_x$ was ~745%. With the increase of vibration frequency, storage modulus (G') and loss modulus (G") of eT-patch were boosted (Supplementary Fig. 9). As the increase of frequency, the value of G" is higher than that of G' at higher frequencies, which behaves as a sticky

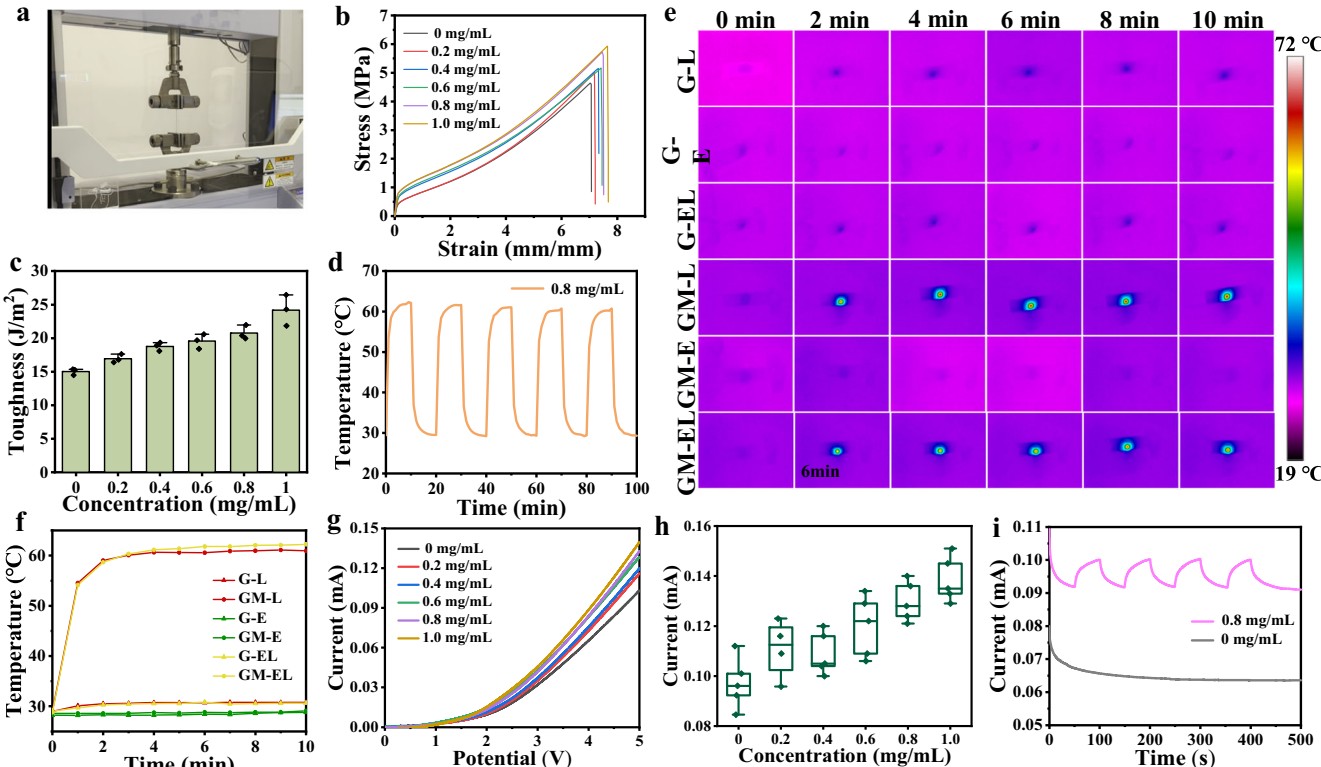

**Fig. 3 | eT-patch performance measurement. a** Photograph of an ionic gel patch during stretching. Typical stress-strain curves (**b**) and toughness (**c**) of the patch doped with different concentrations of MXene (*n* = 3). **d** Cycling temperature profile for the eT-patch doped with 0.8 mg/mL MXene under 808 nm laser irradiation at 0.5 W/cm². Thermal images (**e**) and temperature versus time curves (**f**) for the eT-patch and pure ionic gel patches without MXene doping under Laser, ES, Laser and ES simultaneous stimulation, respectively. Current-voltage curves (**g**) and the corresponding current values (**h**) at 5 V of ionic gel patches doped with MXene of different concentrations (*n* = 5). **i** Current-time curves for the eT-patches and pure ionic gel patches under the off/on of 808 nm laser irradiation. Data are presented as mean ± SD.

material[31]. Remarkably, the photothermal conversion performance of the patches was significantly improved after MXene doping and the temperature of patches was gradually exalted with concentration of MXene rising under 808 nm laser irradiation for 10 min (Supplementary Fig. 10a). Meanwhile, the temperature of patches was apparently elevated with increasing the laser power and thickness of patches (Supplementary Fig. 10b, c). Attractively, temperature measured at different regions of a same patch was basically the same (Supplementary Fig. 10d), which implies that the MXene doping within ionic gel patch is basically uniform. In the subsequent experiments, the ionic gel patch doped with MXene at 0.8 mg/mL was used, the temperature of which is up to about 62.4 °C after 808 nm laser irradiation for 10 min at 0.5 W/cm². Notably, the patch displayed superior thermal stability even if it was illuminated by an 808 nm laser for 10 min each time for five cycles (Fig. 3d). Moreover, the temperature variations of different groups (MXene doping in gels (GM) and pure gels (G), with electrical stimulation (E), laser irradiation (L), and simultaneous treatment of ES and laser (EL), respectively) were monitored through thermal imaging and plotted into curves (Fig. 3e, f). The results demonstrated that the temperature of pure ionic gel patch did not change significantly after the laser and ES, where ES had a little effect on the temperature conversion of the ionic gel patch doped with MXene, while the temperature of ionic gel patch doped with MXene increased evidently with irradiation time lengthened.

The fabricated patch has a good conductivity even if it is twisted, as current passing through it can successfully light up a commercial light-emitting diode (Supplementary Fig. 11a, b). The current-voltage curves of patches doped with different concentrations of MXene were then measured using an electrochemical workstation (Fig. 3g), the current values at 5 V were recorded in parallel experiments several times (Fig. 3h), which manifested that the current values gradually raised with increasing the MXene doping content. We further measured the impedance of the patches to corroborate with their current-voltage behaviors (Supplementary Fig. 12a). The impedance values of the patches before and after doping with MXene were calculated to be about 1539 and 1139 Ω (Supplementary Fig. 12b), respectively. Remarkably, compared with pure ionic gel patch, an apparent change of current was observed from the MXene-doped ionic gel patch (Fig. 3i), as a result of excellent photothermal effect in MXene nanosheets that caused temperature rising to induce heat transfer to the surrounding ionic liquid. It promotes rapid carrier migration and eventually leads to the increase of current.

To prove that the ionic patch developed herein possesses evident advantages over hydrogel patch, the poly (acrylamide-co-acrylic acid) hydrogel doped with MXene, as a control, was prepared and the thermal stability, electrical conductivity and stretching properties of the hydrogel patches were examined. As shown in Fig. S13, the results clearly indicated the poor cycling thermal stability of the hydrogel patch compared with the eT-patch (Supplementary Fig. 13a and Fig. 3d), as evidenced by uncontrollable temperature elevation of the MXene-doped hydrogel patches under intermittent laser irradiation. The ionic gel patches also displayed better electrical conductivity and higher breaking strength and strain than the hydrogel patches (Supplementarys Fig. 13b, c).

Prior to practical biomedical applications, the biocompatibility of the eT-patch was tested. First, we incubated B16F10 cells with extraction medium (2–12 mg/mL) for 24 h, by soaking patches of different quality with culture medium. As shown in Supplementary Fig. 14a, the

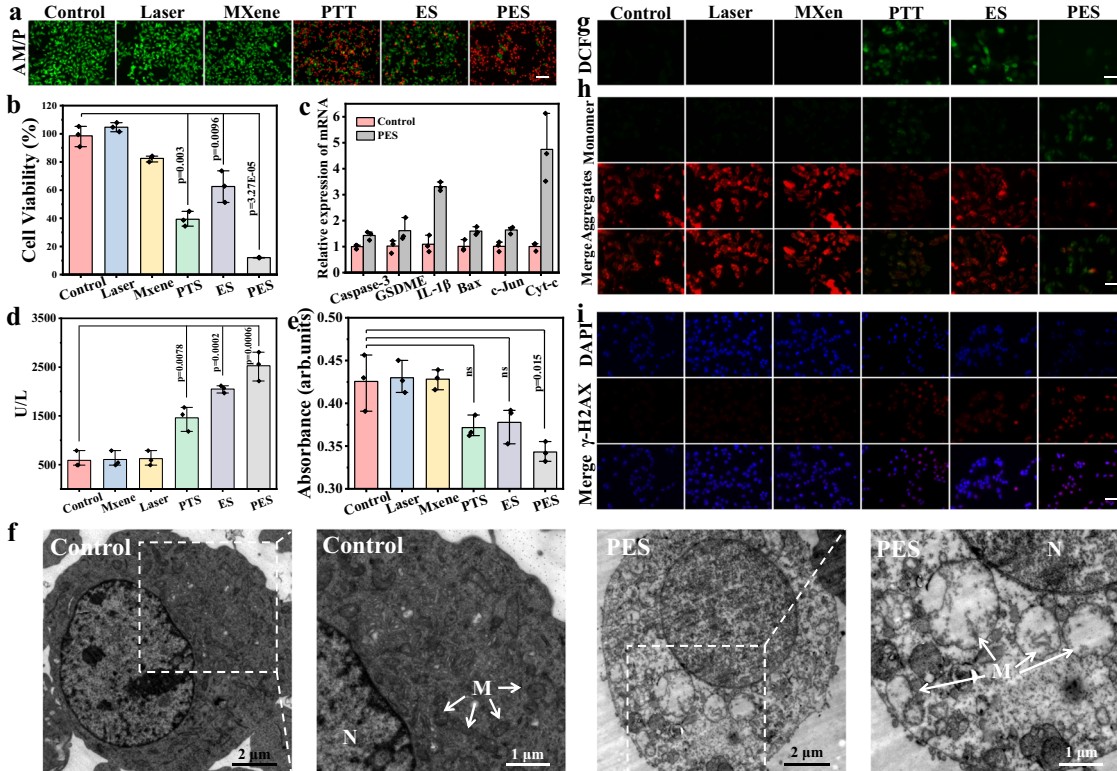

**Fig. 4 | Mechanism study of PES for inducing cell death. a** Live/dead staining fluorescence images of B16F10 cells after the different treatments (Control, Laser, MXene, PTT, ES, and PES). The scale bar is 100 μm. **b** Cell viability of B16F10 cells after the different treatments. *P* values were calculated by two-tailed *t*-tests. (*n* = 3). **c**, RT-PCR analyses of Caspase-3, GSDME, IL-1β, Bax, c-Jun, Cyt-c expression within B16F10 cells before and after the PES treatments *t* (*n* = 4). **d** The lactate dehydrogenase (LDH) release levels from B16F10 cells after the treatments under different conditions. *P* values were calculated by two-tailed *t*-tests. (*n* = 3). **e** ATP content change within B16F10 cells under the same cell number after different treatments tested using the ATP assay kit. *P* values were calculated by two-tailed *t*-

tests. (*n* = 3). **f** Bio-TEM images of B16F10 cells before and after the PES treatments. "M" represents mitochondria, "N" represents cell nucleus. **g** Fluorescence images of ROS within B16F10 cells detected using the DCFH-DA probe after the different treatments. The scale bar is 50 μm. **h** The fluorescence images of the JC-1 monomer and aggregate, and their merged images within B16F10 cells stained by JC-1 after the different treatments. The scale bar is 50 μm. **i** Fluorescence images of DNA double-strand breaks within B16F10 cells after the different treatments (corresponding to red fluorescence of γ-H2AX) followed by DAPI staining (corresponding to blue fluorescence). The scale bar is 50 μm. The experiments were repeated for three times with similar results obtained. Data are presented as mean ± SD.

good cell viability of B16F10 cells was maintained at a concentration of 6 mg/mL of extract medium. We further examined the cell viability of B16F10 cells incubated with higher concentrations of patch extraction medium (10–60 mg/mL) for 10 min followed by continual culture in a normal medium for 24 h. As shown in Supplementary Fig. 14b, the cell survival rates of all tested cells were higher than 80%, which implied good biocompatibility of the patch after treatment for 10 min. To further prove that the eT-patch possesses good biocompatibility at tissue level, the pork tissues stained using hematoxylin-eosin were tested before and after patch pasting (Supplementary Fig. 15). The result indicates that the eT-patches cannot destroy the tissue, which is suitable for potential treatment of skin cancer. Although the suspension of MXene is black, the mixing of MXene with ionic gel becomes transparent due to its optical property change caused by transformation from a colloid state to a gel state. As shown in Supplementary Fig. 16, though the transmittance of patches was gradually reduced with increasing the concentrations of doped MXene, optical transparency of the patches is still good at 0.8 mg /mL of MXene doping. Consequently, the eT-patch prepared can be used to inspect skin response and evaluate treatment effects during the treatment process.

### Mechanisms of cell death during PES treatment
Before applying the eT-patch to skin tumor treatment, we systematically examined and revealed the effects and cell death mechanism of photoelectric stimulation for B16F10 cells. The different currents were firstly applied to treat B16F10 cells cultured on the ITO glasses for

10 min, then the cell viability was tested using the live/dead fluorescent staining and MTT assay (Supplementary Fig. 17), which clearly showed that cell viability was gradually decreased with the stimulation current boosted, indicating that the increasing of current has improved the killing rate of cancer cells. We then chose current value of 5 μA for the subsequent experiments. Subsequently, B16F10 cells were treated with different methods (Fig. 4a, b), the results of which indicated that the cell killing efficacy of PES treatment was dramatically stronger than other treatments. Obviously, laser irradiation in the PES experimental group (applying both electric and photothermal stimulation) had not only an augmented-current effect, but also a photothermal effect, both of which are beneficial for killing cancer cells. In addition, we selected the mouse fibroblasts of L929 cells as the normal cells to check if the method can cause less damage to normal cells. As shown in Supplementary Fig. 18, the tested L929 cells had higher cell survival compared to B16F10 cells. Therefore, the tumor cells were more likely to be damaged during the treatment process. Apparently, we noted that the dead cells induced by ES displayed evident cell swelling which is typical characteristic of pyroptosis[32], whereas the PTT triggered cell death was apoptosis owing to the obvious wrinkling of cell morphology (Supplementary Fig. 19)[33]. Thus, both cell pyroptosis and apoptosis can be triggered during PES treatment, both of which are beneficial to cancer treatments. To validate that the PES can induce cancer cells pyroptosis and apoptosis, the relevant gene expressions before and after the PES treatment were performed using the RT-PCR (reverse transcription polymerase chain reaction). As shown in Fig. 4c, the gene

expressions of markers from pyroptosis (GSDME, Caspase-3, IL-1β) and apoptosis (Bax, c-Jun, Cyt-c) within B16F10 cells were significantly elevated after the PES treatment, compared with control group. To further confirm that pyroptosis was triggered by ES, the lactate dehydrogenase (LDH) as a classical pyroptosis biomarker[34] whose release levels were detected after the different treatments using commercial LDH assay kit. As shown in Fig. 4d, LDH release levels from B16F10 cells in PES and ES groups were palpably higher than that of other groups. Simultaneously, the ATP level within B16F10 cells after the PES treatment was found lower than that under other treatments (Fig. 4e). These results demonstrated that strong pyroptosis of cells was induced by the PES strategy. Furthermore, an obvious mitochondrial destruction with noticeable cavitation and swelling was observed from biological transmission electron microscopy (Bio-TEM) images, as shown in Fig. 4f, which indicates the mitochondrial dysfunction by PES[35]. In addition, the ROS level within B16F10 cells was tested using the 2, 7-Dichlorodihydrofluorescein diacetate (DCFH-DA) which can be hydrolyzed to generate DCFH impermeable membranes, which diffuse internally and be oxidized by non-specific oxygen radicals to generate green fluorescent DCF. As shown in Fig. 4g, distinct green fluorescence imaging of B16F10 cells was observed from PTT and ES groups. However, B16F10 cells showed weak green fluorescence images owing to cell membrane rupture after the laser and electric co-stimulation. Since mitochondrial membrane potential (MMP) is important for maintaining the cellular mitochondrial function and a decrease in MMP is usually associated with the accumulation of ROS, the MMP within cells was examined after the different treatments using JC-1 assay kit (Fig. 4h), the results of which affirmed that the MMP within cells after PES treatment was significantly reduced, resulting in the disappearance of red fluorescence and generation of green fluorescence. As known that ROS level elevating can induce DNA damage in cells[36], we then used γ-H2AX immunofluorescence staining to investigate the possible DNA double-strand breaks that may occur after different treatments. As shown in Fig. 4i, the PES treatment caused the highest degree of DNA damage in test groups (PTT, ES, and PES), which indirectly affirmed the generation of higher ROS levels from PES treatment, and that the ROS produced can severely damage DNA in the nucleus.

## Therapeutic efficacy evaluation of eT-patch for melanoma treatment

To evaluate the treatment effect of the eT-patch for skin tumor under PES, the B16F10 tumor-bearing C57BL/6J mice model was established through subcutaneous transplantation of tumor cells and the therapeutic protocol is displayed in Fig. 5a. The tumor-bearing mice with a volume of about $100\,mm^3$ were first randomly divided into five groups including control, Laser, PTT, ES, and PES groups. Subsequently, the mice were treated with different methods for 10 min per treatment under two consecutive days, and then observed for 13 days. The treatment effect of different methods was estimated after 15 days. To reveal where the current flows, the current values flowing through the eT-patch before and after its covering on the tumor were detected and calculated, as shown in Fig. S20. The current value passing through the free-standing patch (denoted as $I_0$) was calculated to be about $9.3 \times 10^{-5}$ A at 5 V. Notably, the current value ($I_1$) recorded at 5 V was found increased to ~$1.7 \times 10^{-4}$ A after the patch attaching to the tumor, indicating that there is a parallel resistance relationship between the tumor and the eT-patch. An electric current can flow through the superficial layer of the tumor via the patch, as depicted in Fig. S20b. For better understanding, we drew the circuit diagrams corresponding to each of the two cases (Supplementary Fig. 20c, d), and the direction of current flow during the treatment is also labeled correspondingly. In order to estimate the effective depth of the treatment, tumor tissues of mice after the PES treatment were taken for H&E and TUNEL staining. As seen from Supplementary Fig. 21, the effective depth of the method

for tumor ablation can reach ~5 mm. Specially, the real-time infrared thermal images of tumor-bearing mice in the PTT and PES groups displayed a rapid increase of local temperature during the treatment process, which implies that the PTT and PES treatments are well suitable for tumor elimination (Fig. 5b). The thermal imaging of tumor and temperature profiles focused on tumor and the surrounded area (Supplementary Fig. 22) showed that the temperature of tumor surrounded area is about $40\,°C$, which is less possible to damage the surrounding normal tissues during tumor treatment. The tumor sizes of mice after different treatments were recorded through optical photographs (Fig. 5c), which proved that significant suppression effect was enhanced under the PES treatment. The tumor tissues after the different treatments were dissected and weighed, and as shown in Fig. 5d, e, Supplementary Fig. 23 (The weights of mice body and tumors in different groups can be found at Supplementary Table 2), the eT-patch has a superior anti-tumor effect. The average suppression rate based on relative tumor volume was calculated to be ~17.80, 17.70, 14.13, 14.98, and 1.96, respectively, as shown in Fig. 5f (The absolute tumor volumes in different groups can be found at Supplementary Table 3). And the body weight of mice under different groups has no significant loss (Fig. 5g), demonstrating low systemic toxicity of the materials and method. To better evaluate the treatment efficacy, further hematoxylin and eosin (H&E) and Ki-67 staining were performed, and the results of which showed that PES therapy with eT-patch has the most significant tumor damage and proliferation inhibition (Fig. 5h). To further check the damage degree of the tumor after PES treatment, the bio-TEM images were conducted to observe morphology of cells from tumor site. As shown in Supplementary Fig. 24, the organelles of mitochondria were seriously destroyed after PES, compared with control group. Moreover, TUNEL immunofluorescence images of tumor tissues revealed that the PTT and PES therapies of eT-patch obtained the highest level of cancer cell apoptosis. Note that caspase-3 as a pyroptosis marker can be activated to cut GSDME protein to trigger cell pyroptosis[37], as shown in Fig. 5h, the GSDME protein expression level was significantly reduced in the ES and PES groups because more caspase-3 were activated after the ES to cut GSDME. Clearly, both cell apoptosis and pyroptosis were triggered under the PES treatment of the eT-patch, both of which will benefit to the anti-tumor efficacy of the method. Attractively, the survival rate of tumor-bearing mice was greatly improved after the PES treatment (Supplementary Fig. 25). The metastasis of tumor after the treatment was also evaluated through H&E imaging. As shown in Supplementary Fig. 26, no nodules were observed in lungs of mice of all groups, indicating that the treatment had no promoting effect on the metastasis of cancer.

Subsequently, the normal tissues around the tumor in the PES group were checked under different days using the H&E staining. As shown in Supplementary Fig. 27a, the normal tissue around the tumor does not damage during PTT process. Although the tissue just above the tumor was destroyed, the skin and muscle tissues of the mice had returned to normal after 15 days, as seen from H&E staining images (Supplementary Fig. 27b). The major organs of mice after the different treatments were also checked by H&E staining (Supplementary Fig. 28). The result manifested that eT-patch PES treatment has no obvious pathological changes in mice. In addition, there were no significant abnormalities in the routine blood tests of hematological biomarkers in each treatment group compared with the control group, and clustering analysis could not distinguish them completely (Supplementary Figs. 29, 30). All the above results indicated that the PES treatment of cancer using eT-patch has excellent biological safety.

## Discussion

In summary, we developed a wearable and dual-responsive eT-patch composed of ionic gel doped with MXene, which were applied for efficient melanoma treatment by photothermal and electrical

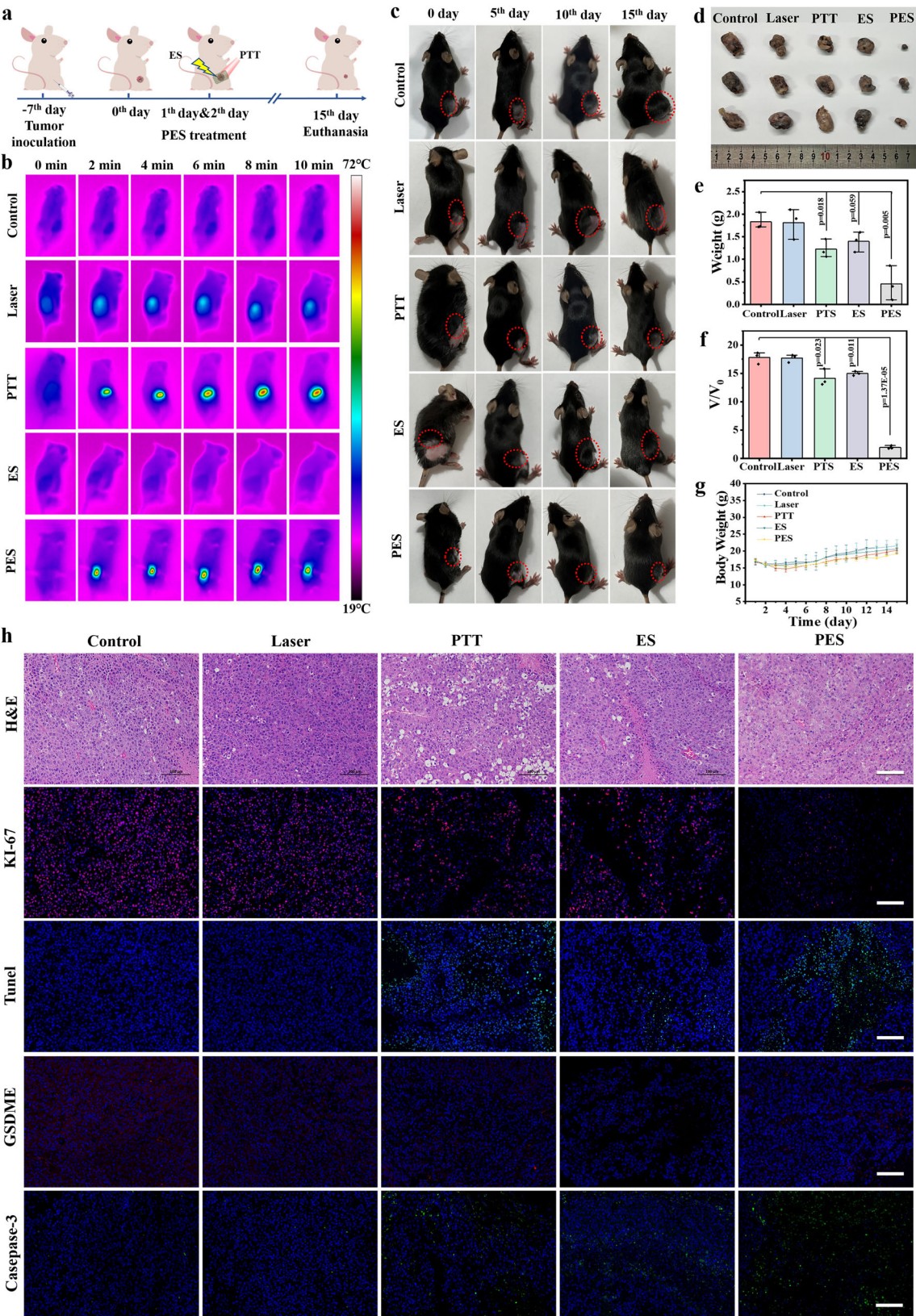

**Fig. 5 | In vivo treatment with melanoma model. a** Schematic diagram of B16F10 tumor-bearing mice model establishment and therapeutic procedures. **b** Thermal images of tumor-bearing mice under different treatments during different time. **c** Representative photographs of the tumor-bearing mice during the various treatment procedures under different days (5th, 10th, and 15th days). The red circle represents melanoma. **d** Photographs of tumors dissected from mice at 15th day of each treatment (Control, Laser, PTT, ES, and PES). **e, f** Tumor weights and relative volume of tumors dissected from each group after therapy for 15 days. *P* values were calculated by two-tailed *t*-tests. (*n* = 3). **g** The body weight curves of tumor-bearing mice in different groups. (*n* = 3). **h** H&E, Tunel, Ki-67, GSDME and caspase-3 staining of tumor tissues collected from the corresponding mice at 15th day of each treatment. The scale bar is 100 μm. The experiments were repeated for three times with similar results obtained. The experiments were repeated for three times with similar results obtained. Data are presented as mean ± SD.

stimulation (PES) co-therapy. The good transparency of the eT-patch achieves real-time visually inspecting of skin response and assesses treatment effect of skin tumor. The underlying cellular mechanisms of PES treatment were preliminarily revealed, which was that both apoptosis and pyroptosis were triggered that associating with ROS-induced MMP reduction and DNA damage. Impressively, the PES treatment of skin cancer with the easily fabricated eT-patch possesses high biosafety and stability, fewer side effects, and flexible modulation. This study offers new opportunities to explore patch materials for effective melanoma treatment avoiding surgery risk and will broaden the biomedical applications of ionic gels.

## Methods

### Ethics statement
All mice were raised and cared according to the guidelines on Laboratory Animals of Jilin University. Meanwhile, all animal procedure was approved by the Laboratory Animal Management Committee of Jilin University.

### Synthesis of MXene doped ionic gel patch
It is known that polyacrylamide (hydrogel) has good biocompatibility for biomedical applications. The acrylamide is insoluble in ionic liquids, while pure polyacrylic acid is soluble in ionic liquids. The acrylamide and polyacrylic acid in ionic liquids can randomly copolymer to form ionic gel, which has high fracture strength by dissipating energy through hydrogen bonding. Therefore, the acrylamide and acrylic acid were selected as precursors for the preparation of ionic gel patches in this work. First, 0.54 g of acrylic acid (AA) and 1.5975 g of acrylamide (AAM) were fully dissolved in 2 mL of 1-ethyl-3-methylimidazolium (EMIES) solution to obtain a homogeneous solution. A certain mass of MXene powder was weighted and made to be uniformly dispersed into 1.3 mL of EMIES solution to mix thoroughly. Subsequently, the crosslinker N, N'-Methylenebis (acrylamide) (MBAA, $C_{MBAA} = 0.1$ mol%) and photoinitiator Irgacure 2959 ($C_{I2959} = 0.1$ mol%) were added to obtain the precursor solution. The precursor solution was poured into the mold and then irradiated under ultraviolet light (proximately 55 mW/cm$^2$) for 5 min. Finally, the MXene doped ionic gel patch was obtained for use.

### Cell Culture
The melanoma cells (B16F10) were obtained from the American Type Culture Collection (ATCC, USA). BF16-F10 cells were cultured into 1640 medium with 10% fetal bovine serum (FBS) and 1% penicillin-streptomycin and maintained in a 37 °C and 5% CO$_2$ humidified incubator.

### Biocompatibility evaluation of MXene and ionic gel patch
Typically, the toxicity of ionic gel and MXene was detected using the standardized MTT assay. The B16F10 cells ($1.0 \times 10^5$) were planted into the each well of 96-well plate for incubation. After that, the cells were washed using PBS three times and incubated with different concentrations of MXene dissolved with 1640 complete medium at 37 °C in 5% CO$_2$ for 24 h. Subsequently, the cells were cleaned using the PBS three times and then 10 μL of MTT solution (5 mg/mL) was added into 90 μL of 1640 complete medium to incubate for another 4 h. Finally, the solution of each well was removed and 150 μL of DMSO were added into each well after reaction to dissolve purple formazan crystals. The absorbance of the wells was detected on a microplate reader with a measurement wavelength of 570 nm. Simultaneously, to assess the toxicity of ionic gel patch, the gel extracts were obtained by soaking different masses of gel in the medium for 10 min to obtain different concentrations of gel extracts from 2 to 12 mg/mL. Whereafter, the different concentrations of gel extracts were treated with B16F10 cells for 24 h and detected using the MTT assay based on the above procedures.

### Cell viability of B16-F10 after electrical stimulation under different currents
Briefly, the B16F10 cells were seeded on conductive glass (ITO glass) to incubate in 5% CO$_2$ incubator at 37 °C for 12 h. After that, the cells were washed using PBS three times and treated with different currents (0, 0.5, 1, 5, and 10 μA) under three electrode system for 10 min. The cell electrode was considered as working electrode and the platinum sheet was set as counter electrode. The reference electrode was Ag/AgCl electrode. After ES for cells, the cells were sequentially cultured for 30 min. Subsequently, the cells were stained with live-dead cell assay kit for 20 min and then cleaned using the PBS three times. The fluorescence imaging of cells was observed using the fluorescence microscope under 40× object. Meanwhile, to further estimate the cell viability after ES under different currents, the cells were digested using the tryptase and detected using the MTT assay kit based on the above experimental procedures.

### Cell viability of B16-F10 cells incubated with MXene after laser irradiation under different laser powers
Typically, the B16F10 cells ($1 \times 10^5$ cells) were seeded in each well of a 96-well plate to incubate for 12 h. The cells cleaned were then treated with MXene (45 μg/mL) dissolved into complete medium for 24 h. Lately, the cells were washed to remove the MXene solution and added fresh culture medium to illuminate by 808 nm laser for 10 min under different laser powers (0.25, 0.5, 0.75, and 1.0 W/cm$^2$). Subsequently, the cells were incubated for another 2 h. Finally, the cell viability of B16-F10 cells was detected using the standardized MTT assay.

### Evaluation of treatment effect of B16-F10 cells under different methods
The anticancer efficiency of electrothermal treatment in vitro was evaluated in B16F10 cancer cells. There are six different treatment groups divided: control, laser, MXene, PTT (MXene + laser), ES, and PES (MXene + laser + ES). B16F10 cells ($10^5$ cells) were first seeded on the ITO glasses and incubated for 12 h. Then the cells were washed using the PBS three times and three groups were incubated with culture medium contained with MXene (45 μg/mL) for 24 h. The control group did not receive any treatment. The laser and PTT groups were exposed to the 808 nm laser at a power of 0.5 W/cm$^2$ for 10 min. The ES and PES groups were stimulated with 5 μA current for 10 min, the difference of which was that the PES group was treated using 808 nm laser irradiation along with ES. After the treatment, the cells were sequentially cultured for 30 min. Subsequently, the cells were removed from ITO glasses using trypsin and incubated in 96-well plates for MTT assay. Simultaneously, the fluorescence images of B16F10 cells after the treatments were detected through the live/dead dye staining (Calcein-AM: 2 μmol/L, PI: 8 μmol/L). The cells were then washed using the PBS three times for 15 min and observed using the fluorescence microscope under 20× objective.

### Detection of DNA double-strand breaks
The DNA damage within cells after different treatments was tested using the immunofluorescence. The B16F10 cells were fixed with 4% paraformaldehyde in PBS for 15 min at 4 °C after the different treatments. After that, the cells were washed using the PBS three times and permeabilized by 0.25% Triton X-100 for 15 min to improve the permeability of the cell membrane at room temperature. After washing three times with PBS, cells were blocked with 1% BSA in PBS for 1 h at room temperature and incubated with diluted primar`y antibody (ab81299) at 4 °C overnight. The next day, cells were washed three times with PBS and incubated with diluted Cy3-conjugated secondary antibody in the dark at room temperature for 1 h. Then 2 μg/mL of DAPI was added to stain cell nucleus and the cells were washed three times with PBS and observed under a fluorescence microscope.

### Mitochondrial membrane potential of B16F10 cell after treatments

The MMP within cells were measured using the commercialized assay kit of JC-1. The cells were stained with JC-1 assay kit for 15 min after different treatments. Subsequently, the cells were washed using the PBS by three times. The fluorescence imaging of cells was observed and collected using Leica DMI6000B microscope with a fluorescence detector with 40× objective (EM: 510–540 nm (JC-1 monomer) and EM: 570-620 nm (JC-1 aggregate)).

### Adenosine triphosphate (ATP) detection in B16F10 cells after stimulation

Commercial ATP Assay Kit was used to detect related energy changes of B16F10 cells after different stimulations. After incubating the stimulated cells for 30 min, the cells were cleaned three times with PBS. Subsequently the cells were collected with trypsin solution and the supernatant was removed. Then 200 μL of cell extract solution was added to a centrifuge tube and ultrasound in an ice bath for 1 min. The mixture was then centrifuged at $3420\,g$ at 4 °C for 10 min and the supernatant was removed from the centrifuge tube. The chloroform (50 μL) was then added to the tube and mixed with full oscillations before being centrifuged again at $3420\,g$ at 4 °C for 3 min. The final supernatant was mixed with the working liquid according to the instructions, and the absorption was detected at 340 nm using an enzyme marker.

### Animal experiments

C57BL/6J mice (4 weeks old, female) were purchased from Beijing HFK Biotechnology Ltd (Beijing, China). All mice were raised and cared according to the guidelines on Laboratory Animals of Jilin University. Meanwhile, all animal procedure was approved by the Laboratory Animal Management Committee of Jilin University. The diameter size of tumors cannot exceed 1.5 cm in therapeutic process ; if it exceeds or affects the quality of the animal's survival, the mice should be euthanized. In our work, we observed and recorded the tumor size during the experiment process. On the 15th day, we found that the tumors in the control group and the laser group exceeded the standard and reached the experimental cycle, and according to the regulations of the Jilin University Laboratory Animal Management Committee, we euthanized the mice in a timely manner, in compliance with the regulations. Before and after treatment, the tumor sites of different groups were cleaned with 75% ethanol to avoid infection. The additional tissue damage of mice was disinfected and the living environment of mice is cleaned up and sterilized regularly. In addition, the mice were limited to five per cage and cultured in a sterile, ventilated environment without any treatment after therapy. The living environment of animals was maintained at a temperature of 25 °C and at 40–70% humidity with a 12 h light/dark cycle, with free access to standard food and water. Tumor volume was calculated as follows: tumor volume (mm³) = 1/2 × (tumor length) × (tumor width)².

### In vivo treatment with melanoma model

The tumor model was established by subcutaneous injection with B16F10 cells ($4 \times 10^5$ cells in 100 μL PBS) in the hind part of the mice. After inoculation of tumor cells for 7 days (tumor value ≈ 100 mm³), the mice were randomly divided into five groups ($n = 4$) as follows: (i) control, (ii) laser, (iii) PTT, (iv) ES, (v) PES. For the laser treatment group, tumors in each mouse were exposed to 808 nm laser (0.5 W/cm², 15 min, once) for two consecutive days; and for the PTT experimental group, the eT-patch was applied to the surface of the tumor and the laser focus on the tumor site. For the ES experimental group, the ES was performed with a 100 μA current by direct current stabilized voltage supply through the eT-patch that covered the tumor; while for the PES group, both laser and ES were applied to the eT-patch on the skin surface of melanoma. Tumor surface temperature and

thermal images were monitored and recorded by an infrared thermography system. Tumor size was measured every 2 days with calipers.

### Reporting summary

Further information on research design is available in the Nature Portfolio Reporting Summary linked to this article.

## Data availability

The main data supporting the results in this study are available within the paper and its Supplementary Information. All raw and analyzed datasets generated during the study are available from the corresponding authors upon request.

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

## Acknowledgements

This work was supported by the National Natural Science Foundation of China (grant No. 22004117 (G.H.Q) and 21675146 (Y.D.J.)), Chinese Academy of Sciences for Special Research Assistant Grant (G.H.Q.), the State Key Laboratory of Electrical Analysis Cross-cooperation project (SKLEACIC202003(Y.D.J.)).

## Author contributions

Y.D.J. conceived the project. X.K.J. and G.H.Q. designed the experiment. X.K.J., S.P.H., and X.K.D. participated in material preparation. G.H.Q. carried out cell experiments. X.K.J., J.K., and G.H.Q. carried out cell and mice experiments. X.K.J. collected all the data. X.K.J., G.H.Q., and Y.D.J. analyzed the data. X.K.J., G.H.Q., and Y.D.J. wrote the manuscript. All the authors contributed to the discussion during the whole project. All authors have given approval to the final version of the manuscript.

## Competing interests

The authors declare no competing interests.
