## [Peer Review File · Nature Communications]

Reviewers' Comments:

Reviewer #1:

Remarks to the Author:

In this article, the authors developed a wearable electrostimulation-augmented ionic-gel photothermal patch doped with Mxene, and applied the patch to skin tumor treatment. The authors investigated the anticancer effect of the patch *in vitro* and *in vivo*, and earned good outcome in B16F10-tumor bearing mice. However, the strategy did not show obvious advantages compared to the previously reported work, and the data seem too preliminary to support the conclusion. Thus, I can't recommend this work to be published in Nature Communications at this stage. Some detailed comments are listed below.

1. The authors mentioned the advantages of the ionic patch over hydrogel, and some experimental data should be provided to support this statement.
2. How to promise the patch to cover the whole tumor without affect normal tissue? As shown in Fig.4, the temperature can reach over 60°C after laser irradiation, which will cause the destruction to the skin or other tissues.
3. Most of the recently reported PTT strategies can ablate tumor at 50°C or even lower. Thus, I wonder why the PTS group showed insufficient tumor ablation outcome?
4. The effective depth of the strategy should be carefully investigated.
5. The mechanism of the synergistic antitumor effect of PTS and ES need to be discussed more clearly.
6. As melanoma is a type of highly metastatic cancer, the metastasis of tumor after treatment should be evaluated, and the long-term survival rates of the mice after different treatments should be monitored.
7. The proper noun error of "Mxene" need to be pointed out. From the first MXene report 'Naguib, M.; Kurtoglu, M.; Presser, V.; Lu, J.; Niu, J.; Heon, M.; Hultman, L.; Gogotsi, Y.; Barsoum, M. W. Adv. Mater. 2011, 23, 4248.' published in 2011, nearly all the MXene-related articles were followed the abbreviation 'MXene' for 'transition metal carbides and carbonitrides', rather the 'Mxene'.

Reviewer #2:

Remarks to the Author:

This manuscript reports a development of a transparent and conductive electrothermal patch for skin tumor treatment. The patch is made from an ionic gel mixed with Mxene flakes. The patch is able to provide photothermal treatment to kill tumor cells. It is also conductive to introduce electrical stimulations. They demonstrated both treatments applied together has a significantly improved effect on reducing the tumor size. While this result is somewhat interesting, the paper is poorly prepared. Description and discussion are mostly confusing and the many necessary details are either missing or not sufficiently discussed. The reviewer does not believe this manuscript has necessary quality and scientific rigor for a publication in Nature Communications.

1. There are two figure 1 in the paper. So the labels are completely messed up. It is very hard to follow which one the authors are referring to in the discussion.
2. What is the ionic gel? It is not discussed in the paper at all.
3. Individual Mxene is characterized and briefly discussed. However, the composite is not discussed at all, which should be the focus of this work. What is the structure? How is the distribution of Mxene in the matrix? Is there any agglomeration? Any difference when the concentration of Mxene changes? How to achieve and sustain the uniform distribution of Mxene in the matrix? What is the interface bonding situation?
4. The suspension of Mxene is black. Why mixing Mxene with ionic gel becomes transparent?
5. The photothermal effect is coming from Mxene. How is the photothermal effect change when the Mxene concentration change? Would the amount of ionic gel and thickness make any difference?
6. Labels in the figures are extremely hard to read, which makes it even harder to understand what the paper presents.
7. Electrical stimulation is applied during the measurement. How ES is introduced?
8. The current density change upon illumination is claimed to be photocurrent, but the explanation is indicating due to conductivity change. These are two completely different concepts. Should be

corrected.

9. What are the impedances? No data are given.

10. What are the B16F10 cells? It is not explained in the paper at all. Seems it is a type of tumor cells. What about normal cells? Would this patch also kill normal cells? How to achieve reasonable selectivity of this treatment?

11. What is the ES signal applied in cell and tumor study? Where does the current flow to?

Reviewer #3:

Remarks to the Author:

The present manuscript entitled, 'A wearable electrostimulation-augmented ionic-gel photothermal patch doped with Mxene for skin tumor treatment' by Ju and coworkers, claims development of a Mxene doped gel patch to be placed topically over skin to target skin tumors. Overall a nice study, but lacks significant information for acceptance in Nature Communications. Hence, the manuscript cannot be considered for publication in Nature Communications in present form.

1.The entire manuscript is full of grammatical errors and sentence formation mistakes. For instance, Page 2, line 68-71, doesn't make sense. The manuscript is loaded with many such errors.

2.The figure quality is extremely poor. The legends of the graphs/fig are also not readable.

3.The authors claim development of patch derived of acrylamide, crosslinked with MBAA and photoinitiator: Irgacure 2959. Why acrylamide? Pls specify. In addition, acrylamide can be crosslinked with MBAA via free-radical polymerization using APS/riboflavin, TEMED? Authors, need to specify the reason for opting for UV based polymerization, as the target group is Melanoma tumors. In addition, much work has already been done using such topical patch for photothermal based tumor targeting. Mere doping with Mxenes, as 2D nanomaterial, is not sufficient to impart novelty. Literature exists on use of 2D Graphene as photothermal nanomaterial in hydrogel patch on similar lines. Pls. explain.

4.Viscoelastic measurements are missing.

5.Electron microscopic images revealing the pore microstructure are lacking.

6.The apoptosis/pyroptosis induction mechanism is also not substantially explained. Gene expression studies and TEM images needs to be included to back the claims by cell culture experiments.

7.Fig. 4 highlighting the in vivo treatment on melanoma models, is not clear and descriptive.

We thank the three referees very much for constructive comments and criticisms, which are very valuable in improving the quality of our manuscript. We have performed additional experiments, and added some important data and carefully revised our manuscript thoroughly to fully address the all concerns raised by the three referees. We reported in this study a new patch material based on *ionogels*, along with a simple and effective method, for preparing a high-performance patch towards electrostimulation-augmented photothermal tumor treatment, with distinct advantages over previously reported patches based mainly on *hydrogels* in the field. We believe that our work is timely and is of sufficient novelty, and the carefully and thoroughly revised manuscript warrants further consideration in Nature Communications.

The point-by-point responses to the referees' comments are addressed below.

Letter to Reviewer #1:

(Remarks to the Author):

In this article, the authors developed a wearable electrostimulation-augmented ionic-gel photothermal patch doped with MXene, and applied the patch to skin tumor treatment. The authors investigated the anticancer effect of the patch in vitro and in vivo, and earned good outcome in B16F10-tumor bearing mice. However, the strategy did not show obvious advantages compared to the previously reported work, and the data seem too preliminary to support the conclusion. Thus, I can't recommend this work to be published in Nature Communications at this stage. Some detailed comments are listed below.

General reply: We thank the reviewer for his/her pertinent comments and very professional suggestions which are very constructive to improve the quality of the manuscript. We have performed additional experiments, and added some important data and carefully revised our manuscript thoroughly to meet the requirement for publication in Nature Communications.

1. The authors mentioned the advantages of the ionic patch over hydrogel, and some experimental data should be provided to support this statement.

Reply: As you suggested, to prove that the ionic patch developed herein possesses evident advantages over hydrogel patch, the poly (acrylamide-co-acrylic acid) hydrogel doped with MXene, as a control, was prepared. The thermal stability, electrical conductivity and stretching properties of the hydrogel patches doped with MXene were estimated, as shown in Figure S13. The results clearly indicated the poor cycling thermal stability of the hydrogel patch compared with the eT-patch (*cf.* Figure S13a & Figure 3d), as evidenced by uncontrollable temperature elevation of the (control) hydrogel patches doped with MXene under intermittent laser irradiation. The

ionic gel patches also displayed better electrical conductivity and higher breaking strength and strain than the hydrogel patches (as shown in Figure S13, b&c). We have added some descriptions about this. Please find the added Yellow-Highlighted sentences in the middle of page 7 of the revised manuscript.

2. How to promise the patch to cover the whole tumor without affect normal tissue? As shown in Fig.4, the temperature can reach over 60 °C after laser irradiation, which will cause the destruction to the skin or other tissues.

Reply: In this work, the patches with comparable size were covered on the whole tumor, which had litter effect for the normal tissue. Experimentally, the laser spots were focused on the tumor site, which can produce the temperature over 60 °C to destroy the tumor. However, the normal tissue around the tumor didn't damage severely during the photothermal treatment process, as shown in Figure S24a. Although the tissue just above the tumor was destroyed, the skin and muscle tissues of the mice had returned to normal after 15 days, as seen from H&E staining images (Figure S24b). In addition, we had found that the normal cells are more tolerant than cancer cells under the same electrical stimulation conditions in our work reported, previously (Anal. Chem. 2019, 91, 1408-1415; Anal. Chem. 2022, 94, 14931-14937). To further prove this, we used L929 cells as normal cells and explored the survival of normal cells under the same stimulation conditions by AM/PI staining and MTT assay (Figure S18). It implied that L929 cells had higher cell viability than B16F10 cells. All these results fully proved that the tumor tissue was more likely to be damaged during the treatment process. We have added some descriptions in the revised manuscript. Please find the added Yellow-Highlighted sentences in the middle of page 8.

3. Most of the recently reported PTT strategies can ablate tumor at 50 °C or even lower. Thus, I wonder why the PTS group showed insufficient tumor ablation outcome?

Reply: Most of the recently reported PTT strategies for tumor ablation were performed completely in a different way, by injecting nanoparticles into the tumor or loaded in the tumor by targeting, and sometimes treated multiply times. (Adv.Mater.2018, 30, 1705980; ACS Nano 2022, 16, 10711–10728; Nature Communications 2022, 13, 5127). That's why other photothermal treatments have good results.

4. The effective depth of the strategy should be carefully investigated.

Reply: As you suggested, the effective depths of the method were investigated by H&E staining imaging of tumor tissue (100 mm³) at different depths after PES treatment at next day. As seen from Figure S20, the effective depth of the method for tumor ablation can reach ~ 5 mm. We have added this useful information in the revision. Please find the Yellow-Highlighted sentences in the middle of page 10.

5. The mechanism of the synergistic antitumor effect of PTS and ES need to be discussed more clearly.

Reply: For better understanding, we have used the term photothermal treatment (PTT)

instead of photothermal stimulation (PTS) in the revised manuscript. As we known, the PTT is a relatively mature cancer treatment technique that relies on photosensitizers to absorb incident light and convert the absorbed photon energy into heat, resulting in a rapid increase of local cell temperature over a certain period of time and the destruction of tumors at high temperatures (Adv. Mater. 2018, 30, 1706320; Adv. Mater. 2018, 30, 1705980; Nano Today 2021, 37, 101073). While electrical stimulation (ES) has the advantages of low cost, simple operation, and not limited by tumor type, and can be applied for cancer treatment by inhibiting cancerous cell proliferation or destroying; and ES induces mitochondrial dysfunction, which ultimately leads to oxygen radical storm production, which leads to disruption of cellular redox homeostasis and DNA damage, ultimately leading to cell death. (Nano Energy 2021 87 106208; Chem. Soc. Rev., 2023, 52, 30–46; Nano Energy 2022, 100, 107471; Adv. Sci. 2023, 10, 220716). We have added the detailed discussions of synergistic antitumor effect of PTS and ES into the revised manuscript. Please find the Yellow-Highlighted sentences in the middle of page 2.

6. As melanoma is a type of highly metastatic cancer, the metastasis of tumor after treatment should be evaluated, and the long-term survival rates of the mice after different treatments should be monitored.

Reply: We thank the reviewer for this very professional suggestion. We have collected lungs from tumor-bearing 20-day mice to observe the metastasis of tumor from H&E staining (Figure S23), and nodules were not found in lungs of mice of all groups, indicating that the treatment had no promoting effect on the metastasis of cancer. We have added this important information on the top of page 11 of the revised manuscript. In addition, we have added the long-term survival rates of the mice after different treatments in Figure S22, and corresponding description has also been added on the top of page 11 of the revised manuscript.

7. The proper noun error of "Mxene" need to be pointed out. From the first MXene report 'Naguib, M.; Kurtoglu, M.; Presser, V.; Lu, J.; Niu, J.; Heon, M.; Hultman, L.; Gogotsi, Y.; Barsoum, M. W. Adv. Mater. 2011, 23, 4248.' published in 2011, nearly all the MXene-related articles were followed the abbreviation 'MXene' for 'transition metal carbides and carbonitrides', rather the 'Mxene'.

Reply: We are really sorry for our carelessness made. We have corrected “Mxene” to “MXene” in the paper.

Letter to Reviewer #2:

(Remarks to the Author):

This manuscript reports a development of a transparent and conductive electrothermal

patch for skin tumor treatment. The patch is made from an ionic gel mixed with MXene flakes. The patch is able to provide photothermal treatment to kill tumor cells. It is also conductive to introduce electrical stimulations. They demonstrated both treatments applied together has a significantly improved effect on reducing the tumor size. While this result is somewhat interesting, the paper is poorly prepared. Description and discussion are mostly confusing and the many necessary details are either missing or not sufficiently discussed. The reviewer does not believe this manuscript has necessary quality and scientific rigor for a publication in Nature Communications.

General reply: We thank the reviewer for his/her pertinent comments and very professional suggestions. We have performed additional experiments to add some important data and made necessary and sufficient discussions in the revised manuscript. We hope the thoroughly revised manuscript has necessary quality and scientific rigor for publication in Nature Communications.

1. There are two figure 1 in the paper. So the labels are completely messed up. It is very hard to follow which one the authors are referring to in the discussion.

Reply: We are sorry for the mistake. We have corrected it in the revised manuscript.

2. What is the ionic gel? It is not discussed in the paper at all.

Reply: Ionic gels are made of a mixture of polymeric organic polymers and salt electrolyte materials that can be electrolyzed as ions. The structure of ionic gels is similar to that of conventional gels because the polymer molecular chains are interconnected or entangled to form a spatial mesh structure and the structural gaps are filled with ions of anions and cations as the dispersing medium. As you requested, we have added these descriptions (Yellow-highlighted) on the top of page 2 of the revised manuscript.

3. Individual MXene is characterized and briefly discussed. However, the composite is not discussed at all, which should be the focus of this work. What is the structure? How is the distribution of MXene in the matrix? Is there any agglomeration? Any difference when the concentration of MXene changes? How to achieve and sustain the uniform distribution of MXene in the matrix? What is the interface bonding situation?

Reply: We have performed additional experiments and characterization to answer your questions. The interior structure of composite ionic gel was observed by SEM image, as shown in Figure S7, which showed a porous lamellar structure. To examine the (uniform) distribution of MXene in the matrix, different positions of the patch were randomly selected to check the temperature variation during 808 laser irradiation. If the MXene in the matrix was agglomerated, the temperature detected will be quite different in different positions. However, we found that the temperature measured at different positions of a patch was basically the same (Figure S10d), which implied that the MXene doping within ionic gel patch is basically uniform. The concentration change of MXene can influence the electrical conductivity and elevated temperature of eT-patch, as shown in Figure 3h and Figure S10a. The interface bonding situation

of MXene and PAA in ionic gels is mainly hydrogen bonds, which promote energy dissipation in polymer chains (Nano Research 2022, 15(5), 4421–4430).

We have revised our manuscript accordingly to add such important information. Please find the Yellow-Highlighted sentences in the middle and on the bottom of page 5, and in the middle of page 6 of the revision.

4. The suspension of MXene is black. Why mixing MXene with ionic gel becomes transparent?

Reply: Although the suspension of MXene is black, the mixing of MXene with ionic gel becomes transparent due to its optical property change caused by transformation from a colloid state to a gel state. We have revised accordingly the manuscript on the bottom of page 7 (Yellow-Highlighted sentence) to add this explanation. We had also investigated the transparency of the patches doped with different concentrations of MXene solution (Figure S16), the results of which demonstrated that the transmittance of patches was gradually decreased with increasing the doping concentrations of MXene.

5. The photothermal effect is coming from MXene. How is the photothermal effect change when the MXene concentration change? Would the amount of ionic gel and thickness make any difference?

Reply: The photothermal effect of patches doped with different concentrations of MXene was examined, as shown in Figure S10a. The results demonstrated that the temperature of patches was gradually increased with increasing the doping concentrations of MXene. Simultaneously, we have further assessed the temperature variation of patches with different thickness but doped with same concentration of MXene (0.8 mg/mL), the results of which showed that the temperature of patches was gradually raised with increasing the thickness, as shown in Figure S10c. We have added these descriptions in the middle of page 6, and highlighted them in Yellow.

6. Labels in the figures are extremely hard to read, which makes it even harder to understand what the paper presents.

Reply: We have carefully revised the relevant labels in the figures.

7. Electrical stimulation is applied during the measurement. How ES is introduced?

Reply: In this work, the ES was introduced through the eT-patch covered on the tumor site and applied the current on eT-patch by direct current stabilized voltage supply. We have added this experiment detail into the experimental part in the middle of page 15, and Highlighted in Yellow.

8. The current density change upon illumination is claimed to be photocurrent, but the explanation is indicating due to conductivity change. These are two completely different concepts. Should be corrected.

Reply: We appreciate this very professional suggestion. We have corrected “an apparent photocurrent” to “an apparent change of current”. And the reason of current change has been explained. Please find the Yellow-Highlighted sentence in the middle of page 7 of the revised manuscript.

9. What are the impedances? No data are given.

Reply: As you requested, we further measured the impedance of the patches to corroborate with the current-voltage behavior (Figure S12). The impedance values of the patches before and after doping with MXene were calculated to be about 1139 Ω and 1539 Ω . We have added this data in the middle of page 7 of the revised manuscript.

10. What are the B16F10 cells? It is not explained in the paper at all. Seems it is a type of tumor cells. What about normal cells? Would this patch also kill normal cells? How to achieve reasonable selectivity of this treatment?

Reply: Mouse melanoma cells of B16F10 are spontaneous tumor cells derived from C57BL/6J mice. B16F10 cells have strong proliferation ability as well as aggressiveness and high tumorigenic rate, and are considered to be ideal cells for tumor model construction. We have explained it in the middle of page 4 in the revised manuscript.

Our previously reported work has fully proved that the normal cells are more tolerant than cancer cells under the same electrical stimulation conditions (Anal. Chem. 2019, 91, 1408-1415; Anal. Chem. 2022, 94, 14931–14937; Anal. Chem. 2021, 93, 13624–13631). In addition, the normal cells are also more heat resistant than cancer cells (see for example, Cancer Immunol Immunother 2003 52: 80-88, Progress in Materials Science 2019, 99, 1–26; Journal of Controlled Release 2020, 328, 59–77). To further prove this, we selected the mouse fibroblasts of L929 cells as normal cells to check if the method will cause more severe damage to the cancer cells (than normal cells) in this work, as shown in Figure S18. The results demonstrated that L929 cells had higher cell survival rate as compared to B16F10 cells. Therefore, the tumor cells were more likely to be damaged during the treatment process. We have added some descriptions in the revision. Please find the Yellow-Highlighted sentences in the middle of page 8.

11. What is the ES signal applied in cell and tumor study? Where does the current flow to?

Reply: In this work, we used the constant current as the ES signal applied in cell and tumor study; by applying the patch to the skin near the tumor to form a parallel connection and introduce stimulation by a constant current. Detailed description can be found in the middle of page 15 of the revised manuscript.

Letter to Reviewer #3:

(Remarks to the Author):

The present manuscript entitled, ‘A wearable electrostimulation-augmented ionic-gel photothermal patch doped with MXene for skin tumor treatment’ by Ju and coworkers, claims development of a MXene doped gel patch to be placed topically over skin to target skin tumors. Overall a nice study, but lacks significant information for

acceptance in Nature Communications. Hence, the manuscript cannot be considered for publication in Nature Communications in present form.

General reply: We thank the reviewer for his/her positive evaluation of our work. We have performed additional experiments, added important data, and provided some significant information and detailed discussion in the revised manuscript. We hope that this carefully and thoroughly revised manuscript now merits publication in Nature Communications.

1. The entire manuscript is full of grammatical errors and sentence formation mistakes. For instance, Page 2, line 68-71, doesn't make sense. The manuscript is loaded with many such errors.

Reply: We apologize for the grammatical and sentence formation mistakes we made. We have carefully checked and revised such errors in the revised manuscript.

2. The figure quality is extremely poor. The legends of the graphs/fig are also not readable.

Reply: We have improved the figure quality to make them more readable.

3. The authors claim development of patch derived of acrylamide, crosslinked with MBAA and photoinitiator: Irgacure 2959. Why acrylamide? Pls specify. In addition, acrylamide can be crosslinked with MBAA via free-radical polymerization using APS/riboflavin, TEMED? Authors, need to specify the reason for opting for UV based polymerization, as the target group is Melanoma tumors. In addition, much work has already been done using such topical patch for photothermal based tumor targeting. Mere doping with MXenes, as 2D nanomaterial, is not sufficient to impart novelty. Literature exists on use of 2D Graphene as photothermal nanomaterial in hydrogel patch on similar lines. Pls. explain.

Reply: It is known that the hydrogel of polyacrylamide has good biocompatibility for biomedical applications (Adv. Mater.2022, 34, 2109764; Nano Energy 2023, 109, 108324). The acrylamide is insoluble in ionic liquids, while pure polyacrylic acid is soluble in ionic liquids. The acrylamide and polyacrylic acid in ionic liquids can randomly copolymer to form ionic gel, which has high fracture strength by dissipating energy through hydrogen bonding. Therefore, the acrylamide was selected in this work. We have added this information on the bottom of page 12 of the revised manuscript.

The preparation of gels by ammonium persulphate (*cf.* Adv. Mater. 2022, 34, 220537) takes more time as well as complex steps compared to our method developed in this study. We used the UV light irradiation to prepare the ionic gel patch, the time of which only take 5 min. In addition, the whole preparation process is relatively simple. In our work, the preparation process of the ionic gel patch is separated from the treatment process of melanoma tumors. The use of UV irradiation for patch preparation will not affect its use for melanoma tumors treatment by PES.

We reported in this study a new material, along with a simple and effective method for preparing a high-performance patch towards ES-augmented photothermal tumor treatment, with advantages over previous reports in the field. In our opinion, the work is of sufficient novelty and the thoroughly revised manuscript merits publication in

Nature Communications.

4. Viscoelastic measurements are missing.

Reply: As you suggested, we have performed additional experiments and added the viscoelasticity data of eT-patch in Figure S9. We have revised accordingly our manuscript to add some descriptions (see Yellow-Highlighted sentences on the top of page 6).

5. Electron microscopic images revealing the pore microstructure are lacking.

Reply: The pore microstructure of eT-patch was provided by SEM imaging, as shown in Figure S7. And we have revised our manuscript accordingly (see Yellow-Highlighted sentences on the bottom of page 5).

6. The apoptosis/pyroptosis induction mechanism is also not substantially explained. Gene expression studies and TEM images needs to be included to back the claims by cell culture experiments.

Reply: We thank the reviewer for his/her very valuable and constructive comment. As you requested, we detected the pyroptosis and apoptosis genes after stimulation by real-time quantitative polymerase chain reaction experiment (Figure 4c), the results of which demonstrated that the apoptosis/pyroptosis can be triggered after the PES treatment. In addition, the cell morphologies before and after PES treatment were further investigated by Bio-TEM (Figure 4f), which manifested the occurrence of pyroptosis by ES. We have revised correspondingly our manuscript to add these important results. Please find the Yellow-Highlighted sentences in the middle and on the bottom of page 8.

7. Fig. 4 highlighting the in vivo treatment on melanoma models, is not clear and descriptive.

Reply: We have revised Figure 4a and added the clear and detailed description of the in vivo treatment on melanoma models. Please find the Yellow-Highlighted sentences in the middle of page 10 of the revised manuscript.

Reviewers' Comments:

Reviewer #1:

Remarks to the Author:

After revision, the authors provided more essential experimental results to support their conclusion, and most of my concerns have been well solved. However, there are still some issues need to be further discussed or investigated before acceptance.

1. To better demonstrate the PTT effect on normal tissue of the strategy, clearer thermal imaging results focused on tumor and the surrounded area are needed to show the temperature variation of tumor tissue and the surrounded normal tissue during treatments.
2. The results in Figure S20 is not clear or sufficient enough to evidence the effective depth, images of the whole tumor sections after H&E and TUNEL-staining need to be provided.

Reviewer #2:

Remarks to the Author:

In the revised manuscript, the authors improved the quality of discussion. However, there still are major issues remaining.

1. There still are two figure 1 and other mislabeled figures, which made the discussion hard to follow.
2. The labels in the figures are still impossible to read. I am not able to tell which result is corresponding to which treatment.
3. How was the ES applied is still not clear. No picture is given. No description about where the current flows. Did the eT-Patch just serve as one electrode? Where the current flow to?
4. The mechanism is unclear. The improved effectiveness is attributed to the combined effects of apoptosis and pyroptosis. However, it is not clear why these two effects would show additive efficacy. The mechanism discussion analysis does not explain how these two effects may boost each other.
5. One related observation from the revision is that the current changes (increases) as the laser is applied. It was suggested that it is because the local heat from MXene made the ionic gel more conductive. This also suggests that the applied current stimulation also increases when laser is applied. The increased efficacy might just be a result of increased electric current? It was showed increased current can also increase the killing rate of cancer cells.
6. One important inconsistency: authors showed the ionic gel with MXene had a higher resistance 1539 ohm. However, the current measured from the MXene doped ionic gel was much higher than that measured from pure ionic gel under the same voltage (Figure 2i), although I can't read the numbers in the figure.

In general, the revised manuscript is still defective with critical inconsistencies. I do not support the publication of this manuscript.

Reviewer #3:

Remarks to the Author:

The authors have made attempts to revise the manuscript in light of the comments suggested. The overall quality of experimental backing to the study has improved. The manuscript can be accepted for publication after some minor changes.

- (i) The entire manuscript still requires grammatical check for sentence formation and meaning.
- (ii) Quality of figures, esp. microscopic images can be improved.

Letter to Reviewer #1:

[Comments to Authors]: After revision, the authors provided more essential experimental results to support their conclusion, and most of my concerns have been well solved. However, there are still some issues need to be further discussed or investigated before acceptance.

General Reply: We thank the reviewer for his/her positive comments on our manuscript and suggestions for further revision. The raised issues were carefully considered, and the detailed results and discussion were also added and Yellow-Highlighted in the revised manuscript. We hope the 2nd revised manuscript can now be published on the Nature Communication.

1. To better demonstrate the PTT effect on normal tissue of the strategy, clearer thermal imaging results focused on tumor and the surrounded area are needed to show the temperature variation of tumor tissue and the surrounded normal tissue during treatments.

Reply: Thank you for the very professional suggestion. We have performed additional experiments and provided the data. Clearer thermal imaging results focused on tumor and the surrounded area and the corresponding temperature curves have been provided in Figure S22. We can find that the temperature at the location of the tumor was significantly higher than the surrounded area after laser irradiation, therefore the surrounding normal tissue cannot be severely damaged (Nature Communications 2015 7:10437). We have added some descriptions and Yellow-Highlighted them on the bottom of page 10 of the revised manuscript.

2. The results in Figure S20 is not clear or sufficient enough to evidence the effective depth, images of the whole tumor sections after H&E and TUNEL-staining need to be provided.

Reply: As you requested, we provided the images of the whole tumor sections after treatment and H&E and TUNEL-staining, as shown in Figure S21. The results indicated that the degree of injury was gradually decreased with increasing the treatment depth. We have added some descriptions and Yellow-Highlighted them in the middle of page 10 of the revised manuscript.

Letter to Reviewer #2:

In the revised manuscript, the authors improved the quality of discussion. However, there still are major issues remaining.

General Reply: We thank the reviewer for his/her positive comments and valuable revision suggestions. We have performed additional experiments to add more important data and provided some detailed explanations into the revised manuscript. We hope that the carefully revised manuscript can now meet the requirements for

publication on Nature Communication.

1. There still are two figure 1 and other mislabeled figures, which made the discussion hard to follow.

Reply: We are sorry for the carelessness we made. We have corrected this mistake in the revised manuscript.

2. The labels in the figures are still impossible to read. I am not able to tell which result is corresponding to which treatment.

Reply: We have carefully checked the labels in the figures and made certain revisions to make them more readable. The resolution of images has been improved to 300 dpi to meet the submission requirements. For better readability, we also added the corresponding treatment conditions in the figure captions.

3. How was the ES applied is still not clear. No picture is given. No description about where the current flows. Did the eT-Patch just serve as one electrode? Where the current flow to?

Reply: To answer this, we have given the schematic picture of PES treatment, drew the circuit diagrams and detailed some description to show the current flows. Exactly, the eT-Patch just serves as wearable one-electrode bioelectrode. To reveal where the current flows, the corresponding currents of individual patch and the one covered on the tumor were detected, respectively, as shown in Figure S20. The current value passing through the free-standing patch as (I_0) was calculated to be about 9.3×10^{-5} A at 5 V. Notably, the current value (I_1) recorded at 5 V was found about 1.7×10^{-4} A after the patch attaching to the tumor, which is a significant increase in current, indicating that there is a parallel resistance relationship between the tumor and the eT-patch. An electric current can flow through the superficial layer of the tumor via the patch. For better understanding, we drew the circuit diagrams corresponding to each of the two cases (Figure S20, c&d), and the direction of current flow during the treatment is also labeled correspondingly. Please find the added Yellow-Highlighted sentences in the middle of page 10 of the revised manuscript.

4. The mechanism is unclear. The improved effectiveness is attributed to the combined effects of apoptosis and pyroptosis. However, it is not clear why these two effects would show additive efficacy. The mechanism discussion analysis does not explain how these two effects may boost each other.

Reply: It has known that both pyroptosis and apoptosis of cells have certain anti-tumor effect (Nano Lett. 2021, 21, 8281-8289; Small 2022, 18, 2202161). We are sorry for our unclear expression in page 11 of the manuscript to mislead you. Our results indicated that both the apoptosis and pyroptosis triggered therein are beneficial to kill the tumor. We have corrected the related description (wording) in the revised manuscript (see Yellow-Highlighted sentences in page 11 of the revised manuscript).

5. One related observation from the revision is that the current changes (increases) as the laser is applied. It was suggested that it is because the local heat form MXene made the ionic gel more conductive. This also suggests that the applied current

stimulation also increases when laser is applied. The increased efficacy might just be a result of increased electric current? It was showed increased current can also increase the killing rate of cancer cells.

Reply: The reviewer is correct. The current stimulation applied also increased when laser was applied, as shown in Figure 3i. The two main contributions of increased efficacy for tumor treatment in this study are photothermal treatment and the augmented stimulation current. Attractively, the cell viability was gradually reduced with increasing the stimulating current, as shown in Figure S17b, indicating that the increasing of current do improved the killing rate of cancer cells. Therefore, laser irradiation in the PES experimental group (applying both electric and photothermal stimulation) had not only an augmented-current effect, but also a photothermal effect, both of which are beneficial for killing cancer cells. We have added some related descriptions in the revised manuscript. Please find the Yellow-Highlighted sentences on page 8 of the revised manuscript.

6. One important inconsistency: authors showed the ionic gel with MXene had a higher resistance 1539 ohm. However, the current measured from the MXene doped ionic gel was much higher than that measured from pure ionic gel under the same voltage (Figure 2i), although I can't read the numbers in the figure.

Reply: We apologize for our carelessness in misleading you. In the Figure S12, we can find that the MXene-doped ionic gel has a lower resistance, so it has a higher current. We have carefully revised the manuscript accordingly, as seen from the Yellow-Highlighted sentences in the middle of page 7 of the revised manuscript.

Letter to Reviewer #3:

[Comments to Authors:] The authors have made attempts to revise the manuscript in light of the comments suggested. The overall quality of experimental backing to the study has improved. The manuscript can be accepted for publication after some minor changes.

(i) The entire manuscript still requires grammatical check for sentence formation and meaning.

Reply: We thank the reviewer for his/her positive comments and suggestions. We have carefully checked the grammatical issues throughout the manuscript..

(ii) Quality of figures, esp. microscopic images can be improved.

Reply: As you requested, we have improved the quality of figures and submitted images with a resolution of 300 dpi in accordance with the image requirements of the submission guidelines.

Reviewers' Comments:

Reviewer #1:

Remarks to the Author:

My concerns have been basically solved after the authors' revision. Before publication, I suggest the authors to indicate the direction of the tumor presented in Fig.S21 and the location of the patch should be marked.

Reviewer #2:

Remarks to the Author:

In the revision, the authors addressed most of my previous concerns with good detail. One remaining issue is regarding the animal treatment. While there is a protocol for standard tumor treatments, additional tissue damage was observed from this new treatment approach. However, no specific care was mentioned for this additional tissue damage. Authors should clarify how this observed tissue damage was monitored and regulated to ensure the animal welfare.

Letter to Reviewer #1:

Comments to Authors:

My concerns have been basically solved after the authors' revision. Before publication, I suggest the authors to indicate the direction of the tumor presented in Fig.S21 and the location of the patch should be marked.

Reply: We thank the reviewer for his/her positive suggestion. We have marked the position of the patch and the direction of the tumor in Fig.S21.

Letter to Reviewer #2:

Comments to Authors:

In the revision, the authors addressed most of my previous concerns with good detail. One remaining issue is regarding the animal treatment. While there is a protocol for standard tumor treatments, additional tissue damage was observed from this new treatment approach. However, no specific care was mentioned for this additional tissue damage. Authors should clarify how this observed tissue damage was monitored and regulated to ensure the animal welfare.

Reply: Thank you for the very professional suggestion. In this experiment, the tumor skin tissues of mice after two days treatment were cleaned and disinfected using 75% ethanol. The additional tissue damage of mice was disinfected and the living environment of mice was cleaned up and sterilized regularly, which is helpful to promote skin tissues self-healing. Meanwhile, the skin tissues damage and recovery degrees were monitored using the HE staining after treatment under different days (Fig.S27b). The results have indicated that the inflammation of the skin tissues was eliminated and the damaged skin tissue restored health under the careful care. The damage tissue was carefully monitored and regulated to ensure the animal welfare after treatment in this work. We have added some sentences and Yellow- Highlighted in the revised manuscript in the middle of page 16.

Reviewers' Comments:

Reviewer #2:

Remarks to the Author:

The authors addressed all my remaining concerns. The manuscript can be accepted for publication as it.